# Controversies Surrounding Albumin Use in Sepsis: Lessons from Cirrhosis

**DOI:** 10.3390/ijms242417606

**Published:** 2023-12-18

**Authors:** Christian J. Wiedermann

**Affiliations:** 1Institute of General Practice and Public Health, Claudiana—College of Health Professions, 39100 Bolzano, Italy; christian.wiedermann@am-mg.claudiana.bz.it; 2Department of Public Health, Medical Decision Making and HTA, University of Health Sciences, Medical Informatics and Technology-Tyrol, 6060 Hall, Austria

**Keywords:** sepsis, albumin therapy, hypoalbuminemia, volume resuscitation, cirrhosis

## Abstract

This narrative review critically examines the role of albumin in sepsis management and compares it to its well-established application in liver cirrhosis. Albumin, a key plasma protein, is effective in the management of fluid imbalance, circulatory dysfunction, and inflammation-related complications. However, its role in sepsis is more intricate and characterized by ongoing debate and varied results from clinical studies. In sepsis, the potential benefits of albumin include maintaining vascular integrity and modulating inflammation, yet its consistent clinical efficacy is not as definitive as that in cirrhosis. This review evaluated various clinical trials and evidence, highlighting their limitations and providing practical insights for clinicians. It emphasizes identifying sepsis patient subgroups that are most likely to benefit from albumin therapy, particularly exploring the correction of hypoalbuminemia. This condition, which is significantly corrected in patients with cirrhosis, may have similar therapeutic advantages in sepsis. The potential effectiveness of albumin in the low-volume resuscitation and deresuscitation phases of sepsis management was noted. Given the safety concerns observed in cirrhosis, such as pulmonary edema and hypervolemia associated with albumin therapy, cautious integration of albumin into sepsis treatment is mandatory. Personalized albumin therapy is advocated for tailoring strategies to the specific needs of each patient, based on their clinical presentation and underlying conditions. The need for further research to delineate the role of albumin in sepsis pathophysiology is underscored. The review emphasizes the importance of conducting trials to assess the effectiveness of albumin in correcting hypoalbuminemia in sepsis, its impact on patient outcomes, and the establishment of appropriate dosing and administration methods. This approach to albumin use in sepsis management is posited as a way to potentially improve patient outcomes in this complex clinical scenario while being mindful of the lessons learned from its use in cirrhosis.

## 1. Introduction

Albumin, a crucial plasma protein, plays key roles in maintaining oncotic pressure, transporting substances, regulating pH, and participating in various physiological processes [1,2,3,4]. Its diverse functions render it vital in clinical settings for managing conditions, such as cirrhosis [5] and critical care [6]. Understanding the various roles of albumin is important, especially in the context of cirrhosis and sepsis.

Sepsis, a systemic response to infection, is a major cause of death in intensive care units [7,8]. It involves immune dysfunction characterized by simultaneous pro-inflammatory and anti-inflammatory responses, leading to the release of chemokines and pathogen-related molecules [9]. This dysregulation triggers pro-inflammatory reactions followed by inadequate anti-inflammatory responses that affect pathogen clearance and tissue damage. Microangiopathies activate complement and coagulation pathways, exacerbating organ dysfunction. Interactions between molecular mediators, pathogen-associated molecular patterns (PAMPs), and toll-like receptors on antigen-presenting cells initiate an immunological signaling cascade [10]. These combined immunological, microvascular, and coagulation changes lead to organ dysfunction and affect patient outcomes.

The use of albumin in sepsis presents contrasting effects and is controversial. It is considered for volume resuscitation in septic patients because of its plasma volume expansion capability and potential to improve hemodynamic stability [6,11]. Some studies indicate that albumin might be more advantageous than crystalloids, especially in septic shock [12]; however, these benefits have not been consistently confirmed, leading to debates and weak guideline recommendations [11,13,14,15]. Unlike its established role in cirrhosis-related complications [16], the clinical benefits of albumin in sepsis are less definitive.

Concerns regarding albumin in sepsis include the lack of clear benefits, increased costs, and safety issues such as the risks of hypervolemia, pulmonary complications, myocardial insufficiency, and hemorrhagic complications [17]. Albumin may exacerbate lung edema in patients with chronic heart failure [18] and increase bleeding risk due to hemodilution [19], as observed in hospitalized septic patients with cirrhosis [20].

The role of albumin in cirrhosis and sepsis is distinct: it is effective in cirrhosis for managing complications like large-volume paracentesis (LVP), acute kidney injury (AKI), hepatorenal syndrome (HRS)-AKI, and spontaneous bacterial peritonitis (SBP) through volume expansion and anti-inflammatory actions. In sepsis, however, its benefits are less clear due to disease heterogeneity and varying clinical presentations. Identifying the sepsis subgroups that could benefit from albumin is key for effective clinical study design and optimization. This review explores the role of albumin in sepsis based on its efficacy in cirrhosis. A focused literature search was conducted using PubMed and Embase, utilizing key terms such as ‘albumin’, ‘sepsis’, ‘cirrhosis’, and ‘liver disease’. Additionally, key studies’ citation lists on Google Scholar were manually screened to uncover relevant publications and seminal works. The selection of studies was guided by their relevance to the topic and their contribution to understanding the clinical and mechanistic aspects of albumin in these diseases. The narrative review will conduct a comparative analysis, assess clinical trials and evidence, include limitations, and offer practical recommendations. This review aims to emphasize personalized albumin therapy in sepsis and underscores the need for further research to define the role of albumin in sepsis pathophysiology and its potential benefits for specific patient groups.

## 2. Interplay between Liver Function and Gut Microbiota in Sepsis and Cirrhosis

The role of the liver in the pathophysiology of sepsis is linked to its immune functions and interactions with the gut. In cirrhosis, the intestinal microbiota and microbial PAMPs trigger liver inflammation and complications [21]. Hepatocyte decline in sepsis affects metabolism and immune function, as these cells regulate pro- and anti-inflammatory proteins and sequester immune-controlling alarmins [22].

Sepsis disturbs the intestinal barrier, leading to PAMPs and damage-associated molecular patterns (DAMPs) migrating into the liver and body [23]. The functions of the liver in bacterial elimination, detoxification, cytokine release, and the inflammatory response are crucial. Imbalanced immune responses in the liver can impede pathogen clearance and metabolic processes, thereby affecting the intestinal barrier and microbiota balance [21,23].

The effects of albumin on the gut–liver axis have been studied in burn injuries. Hypoalbuminemia, which is common in burns, leads to immunosuppression and susceptibility to infection [24]. A study showed that albumin resuscitation affected burn-induced immunosuppression, bacterial translocation, and gut endotoxin absorption, although the type of solution (albumin or crystalloid) had no significant effect [25]. This indicates that maintaining albumin levels might support the gut–liver axis in sepsis, potentially reducing bacterial translocation and associated complications; however, direct evidence in the context of sepsis is limited.

Given the intricate interplay between liver function and the gut microbiota in both sepsis and cirrhosis, the theoretical rationale for closely comparing albumin therapy in these conditions is evident. This parallel underscores the potential for applying insights from albumin efficacy in cirrhosis to the management of sepsis [26].

## 3. Use of Albumin in Cirrhosis

Cirrhosis, characterized by liver fibrosis and scarring due to chronic liver damage frequently caused by alcohol, viral hepatitis, nonalcoholic fatty liver disease, or autoimmune diseases, leads to various complications. Patients with cirrhosis, especially those with decompensation or acute-on-chronic liver failure, often require intensive care and have high in-hospital mortality rates. AKI is a common complication typically managed with albumin [27,28]. The complexity of cirrhosis affects multiple organs, necessitating a comprehensive system-based approach to ICU care [29].

Albumin is crucial in managing cirrhosis-related complications, such as ascites, HRS, AKI, and SBP [30]. Insights from the effectiveness of albumin in cirrhosis could inform its use in other conditions, including sepsis.

Caraceni et al. [31] stressed the need for interventions targeting the key pathophysiological aspects of cirrhosis decompensation, such as portal hypertension, gut microbial translocation, and systemic inflammation. Human albumin has been identified as a potential therapeutic candidate. Cirrhosis, marked by portal hypertension and immune dysfunction, leads to systemic inflammation and oxidative stress, and is driven by DAMPs and PAMPs. This inflammation, along with vasodilation and myocardial dysfunction, causes effective hypovolemia and circulatory dysfunction, escalating to multiorgan dysfunction, higher morbidity, and mortality. Albumin administration is potentially effective in managing cirrhosis-related complications, including immune and circulatory dysfunctions [31] (Figure 1).

### 3.1. Effectiveness of Albumin Therapy in Various Cirrhosis Complications

The role of albumin in cirrhosis extends beyond its oncotic functions, such as plasma volume expansion, to diverse non-oncotic properties, including substance binding, antioxidant actions, inflammation modulation, immune response, cardiac function enhancement, and endothelial integrity restoration [30,32]. It is used to manage post-paracentesis circulatory dysfunction, SBP, AKI, and HRS alongside vasoconstrictors, as per international guidelines [16].

A recent umbrella review of meta-analyses [33] indicated that high-dose, short-, and long-term albumin infusion significantly reduced mortality in patients with cirrhosis. However, its impact varies: long-term infusion reduces ascites recurrence but not mortality; in large-volume paracentesis (LVP), it lowers PPCD and hyponatremia without affecting mortality or renal impairment; for overt hepatic encephalopathy (HE), albumin improves severity but not mortality; in SBP, albumin reduces mortality and renal impairment risk, a benefit not seen in non-SBP infections; type-1 HRS patients may experience increased survival with each 100 g cumulative albumin dose, but without HRS reversal. This review confirms albumin’s mortality reduction effectiveness in SBP and HRS, emphasizing the importance of albumin dose, duration, and disease severity in its efficacy [33].

### 3.2. Circumstances Where Albumin Therapy in Cirrhosis Shows Limited Effectiveness

Recent studies of human albumin in decompensated cirrhosis have yielded mixed results. The “human Albumin for the treatmeNt of aScites in patients With hEpatic cirrhosis” (ANSWER) study [34] showed that human albumin significantly improved 18-month overall survival, eased ascites management, and reduced severe complications compared to standard medical treatment. However, the “Midodrine-Albumin in Cirrhotic patients awaiting liver transplantation” (MACHT) trial [35] found no significant differences in complications or mortality in patients treated with human albumin. Similarly, the “Albumin To prevenT Infection in chronic liveR failure” (ATTIRE) study [20] did not show improvements in infection, renal dysfunction, or mortality in patients hospitalized with an acute decompensation of cirrhosis.

These varying outcomes suggest that the effectiveness of human albumin depends on factors such as patient characteristics, study duration, and the dosage and timing of albumin administration, which were all different. While short-term administration has been effective in specific complications, such as SBP and HRD-AKI, long-term treatment may benefit patients with stable decompensated cirrhosis and grade 2 or 3 uncomplicated ascites, especially if it adequately increases serum albumin levels (see below).

### 3.3. Insights from Albumin Therapy in Cirrhosis for Sepsis Management

When considering the use of albumin infusions in cirrhosis, certain insights emerge from the available data, as summarized in Table 1.

#### 3.3.1. Inconsistent Effectiveness of Albumin Therapy in Diverse Cirrhosis Scenarios

Albumin infusion does not exhibit uniform effectiveness across all forms or complications of cirrhosis. Albumin infusion was found to be effective in reducing mortality in cirrhotic patients with SBP and type-1 HRS. In contrast, albumin did not demonstrate a significant beneficial effect on overall survival in long-term infusion for ascites recurrence in patients with cirrhosis [35]. Although albumin infusion reduced the incidence of post-paracentesis circulatory dysfunction and hyponatremia, it did not affect overall mortality or renal impairment in cirrhotic patients undergoing LVP. Infusion improved the severity of overt hepatic encephalopathy but did not influence overall mortality in cirrhotic patients with this complication.

These findings suggest that the effectiveness of albumin infusion in reducing mortality varies across different clinical scenarios of cirrhosis. It may be particularly beneficial in high-dose regimens, SBP, and type-1 HRS, while its impact on survival is less pronounced in cirrhotic patients with ascites, those undergoing LVP, or those experiencing overt HE [33]. The effectiveness scenarios align closely with the colloid properties of albumin, focusing on fluid redistribution.

#### 3.3.2. Organ Dysfunction Conditions Demanding Volume Expansion

Albumin is more effective in situations in which volume expansion is deemed necessary, particularly when the pathophysiology is characterized by systemic inflammation and circulatory organ dysfunction. Its beneficial role in SBP and AKI-HRS is tied to its rapid action in volume expansion and the need for substantial quantities administered within a short timeframe. Notably, it excels in treating SBP, a complication commonly associated with the development of AKI and sepsis in cirrhotic patients [36,37]. When combined with antibiotics, albumin significantly reduces the risk of renal failure and overall mortality compared with antibiotics alone [38]. 

Another essential application of albumin is in the management of AKI, which is prevalent in decompensated cirrhosis. HRS, which is unique to cirrhotic patients, significantly contributes to AKI. Diagnosis often follows a diagnostic algorithm that includes high-dose albumin administration. Patients with specific AKI stages or non-responders may benefit from the combined effects of albumin and terlipressin, as supported by various trials [39], including those on acute-on-chronic liver failure and paracentesis-associated AKI [40].

#### 3.3.3. Disease Severity and Hypoalbuminemia

Disease severity, often linked to hypoalbuminemia, influences albumin dose requirements in the liver. Prognostic scoring systems such as Child–Pugh [41], ALBI [42], and GLOBE [43] include serum albumin levels. Guidelines recommend short-term, high-dose albumin for critical illness; however, its necessity varies based on liver and renal function [44]. In decompensated cirrhosis complications, such as hepatic encephalopathy [43] and non-SBP bacterial infections [45,46,47], short-term albumin shows limited clinical or survival benefits.

The ATTIRE study focused on patients with acute or worsening cirrhosis complications and moderate hypoalbuminemia. The protocol targeted maintaining the serum albumin level at 3 g/dL for up to 14 days [20]. The primary endpoints were infection, renal dysfunction, and mortality between days 3 and 15, or until discharge. Secondary endpoints included survival rates at 28 days, 3 months, and 6 months, and incidences of primary endpoint components. No significant differences were noted in the primary or secondary endpoints between the control and albumin groups, suggesting potential underdosing in long-term decompensated cirrhosis studies.

These results emphasize tailoring albumin prescriptions based on disease severity and context in decompensated cirrhosis. Short-term albumin is effective in SBP and HRS when combined with vasoconstrictors.

#### 3.3.4. Dose and Duration of Albumin Administration

Comparative studies of long-term albumin use in decompensated cirrhosis, as observed in the ANSWER [34] and MACHT [34] trials, offer insights [5,31]. The ANSWER trial, which included patients with stable decompensated cirrhosis and grade 2 and 3 ascites (median MELD score~12), showed a 38% decrease in the mortality hazard ratio and better ascites management with albumin supplementation. The MACHT trial, which involved transplant-awaiting patients with ascites and a median MELD score of 17, found no significant differences in complications or mortality over 12 months between the albumin and non-albumin groups. The albumin dose in MACHT was approximately half of that in ANSWER. Serum albumin levels in MACHT remained around 3.1 g/dL, but increased to 3.8 g/dL in ANSWER, where levels over 4 g/dL after one month correlated with over 90% survival at 18 months [48].

The Pilot-PRECIOSA study showed that higher albumin doses, raising serum levels close to 4 g/dL, improved circulatory function and reduced inflammation in decompensated cirrhosis patients [47]. This suggests that specific serum albumin concentrations are crucial for non-oncotic benefits, such as circulatory stability and reduced inflammation, without impacting portal pressure. High-dose albumin significantly lowered the levels of pro-inflammatory cytokines such as interleukin-6 and vascular endothelial growth factor. The INFECIR-2 study investigated the impact of short-term antibiotics alone versus a combination of albumin and antibiotics on cytokine levels in 78 patients with decompensated cirrhosis and bacterial infections [47]. This study reinforced the immunomodulatory effects of albumin, highlighting its role in reducing systemic inflammation and cardiocirculatory dysfunction.

In summary, the effectiveness of albumin in decompensated cirrhosis is heavily influenced by the dose and duration of administration. 

#### 3.3.5. Safety of Albumin Infusion

Careful consideration of albumin dosing and administration duration is essential to mitigate the risk of complications, particularly hypervolemia-induced pulmonary issues, in cases of pre-existing hypervolemia or impaired cardiac pumping function. An increased risk of pulmonary edema was observed in albumin studies of cirrhosis and bacterial infections unrelated to SBP, with a high Charlson comorbidity index [45], additional albumin dose, and pneumonia [20] as possible risk factors.

In HRS-AKI patients with increased risk factors for pulmonary edema, caution is required because of the combined effects of albumin infusion and terlipressin [49].

### 3.4. Summary of Insights and Implications of Albumin Therapy in Cirrhosis Management

Cirrhosis, a disease characterized by liver fibrosis and various complications, often requires intensive care, especially in patients with decompensation or acute-on-chronic liver failure. Albumin plays a critical role in the management of cirrhosis-related complications, including ascites, HRS, AKI, and SBP. This review examines the various applications of albumin in cirrhosis, highlighting its roles beyond volume expansion, such as antioxidant effects and inflammation modulation.

The efficacy of albumin therapy varies in different cirrhosis scenarios. High-dose, short-, and long-term albumin infusions have shown significant mortality reduction in certain conditions such as SBP and type-1 HRS, but albumin’s impact on survival is less evident in other scenarios such as LVP and HE. The review also notes the mixed results of trials such as ANSWER, MACHT, and ATTIRE, and emphasizes the importance of considering patient characteristics, trial duration, and dosage in determining the efficacy of albumin.

It also discusses the role of albumin in conditions requiring volume expansion, particularly in the context of systemic inflammation and circulatory organ dysfunction. The importance of disease severity and hypoalbuminemia in tailoring albumin therapy is also addressed, emphasizing the need for context-specific prescribing.

In terms of safety, the need for careful consideration of albumin dosing and duration is highlighted, particularly to reduce risks such as pulmonary problems in patients with pre-existing conditions. The importance of albumin dose, duration, and specific clinical context for its efficacy in the management of decompensated cirrhosis is reiterated, and insights into its potential implications for the management of sepsis are offered.

## 4. Use of Albumin in Sepsis

Sepsis and septic shock management involve several key elements [14]. Prompt initiation of empirical antimicrobial therapy for microbial testing is critical. Fluid replacement with crystalloids tailored to patient-specific factors, such as fluid tolerance and responsiveness, is crucial. To maintain a mean arterial pressure (MAP) above 65 mmHg, norepinephrine and vasopressin are used in refractory shock. Mechanical ventilation strategies include reduced tidal volume and preventive measures against venous thromboembolism and glycemic control are recommended.

Recent advancements have highlighted the importance of fluid resuscitation in sepsis patients. Traditional large-volume fluid administration [50] is now questioned as it may not improve microcirculation [51] and could worsen endothelial dysfunction [52]. An individualized approach, with repeated boluses of 250–500 mL crystalloids and early vasopressor use if needed, is advised [53]. Aligning with these insights, the SSC 2021 guidelines emphasize continuous assessment for guiding fluid therapy to avoid fluid overload in sepsis patients. Techniques such as inferior vena cava echocardiographic assessment [54] and advanced hemodynamic monitoring [55] are recommended for the real-time evaluation of fluid responsiveness. These methods help in tailoring fluid administration more precisely. The “Crystalloid Liberal Or Vasopressors Early Resuscitation in Sepsis” (CLOVERS) trial found no significant difference in 90-day mortality between restrictive and liberal fluid strategies [56], but its generalizability is debated [57]. Patients in the restrictive fluids group still received, in the first 24 h of treatment, a median of 3300 mL of crystalloid vs. a median 5400 mL received by the liberal fluids group, as the protocol deviated from that recommended in the SSC guidelines [11,57]. In the CLOVERS trial, the mortality in patients with end-stage renal disease was notably higher in the restrictive group, and the trial was stopped early for futility.

Further research is needed to understand the optimal fluid administration for sepsis, especially in different economic settings [58] and comorbid patient characteristics [18]. The choice and volume of resuscitation fluids remain unanswered [59]. In cirrhosis, where the benefits of albumin are clearer, the role of fluid choice, including hyper-oncotic albumin for small-volume resuscitation and hypoalbuminemia correction, is important.

### 4.1. Clinical Trials and Evidence

In sepsis management, crystalloids are preferred for fluid resuscitation, with balanced solutions increasingly used in early septic shock owing to concerns about chloride-induced hyperchloremia. The role of albumin in early resuscitation is debated, showing evidence of a moderate mortality benefit [60]. NICE recommends crystalloids as a first-line treatment for hypotension in sepsis, considering albumin as a second-line therapy in severe cases [15]. The Surviving Sepsis Campaign suggests albumin levels in patients needing large crystalloid volumes [11].

Fluid overload poses a risk for sepsis management [61]. Restrictive resuscitation and timely fluid removal, with an individualized approach that considers the clinical context and comorbidities, are important. There is growing interest in the potential of albumin for small-volume resuscitation, especially in diuretic-resistant cases and kidney replacement therapy [59,62]. However, a consensus is lacking regarding the concentration of albumin solution, administration volumes, and their combination with other fluids [60].

#### 4.1.1. Randomized Controlled Trials of Albumin in Sepsis-Induced Hypotension

Several randomized controlled trials have been conducted over the last two decades, and they are summarized in Table 2.

The “Saline versus Albumin Fluid Evaluation” (SAFE) trial [63] in 2004 compared 4% albumin solution and normal saline in critically ill ICU patients, and found similar mortality, organ failure, and ICU stay outcomes at 28 days. However, a post hoc analysis suggested that albumin might reduce death odds in severe sepsis cases [64].

The “Early Albumin Resuscitation in Septic Shock” (EARSS) study [65] evaluated early hyper-oncotic albumin use in patients with septic shock. No significant difference in mortality was found between the albumin and saline groups, with similar outcomes in organ dysfunction, ICU/hospital stays, and renal and pulmonary tolerance.

The “Albumin Italian Outcome Sepsis” (ALBIOS) trial investigated 20% albumin with crystalloids versus crystalloids alone in patients with sepsis or septic shock, showing similar 90-day survival rates [66]. A subgroup analysis indicated a potential mortality reduction with albumin in septic shock cases [67].

The “lactated Ringer versus Albumin in early Sepsis theraPy” (RASP) trial [68] compared 4% albumin solution and lactated Ringer’s in septic cancer patients, finding no significant mortality differences at 7 or 28 days.

In the “Fluid Resuscitation In Sepsis-induced hypotension among patients with Cirrhosis” (FRISC) trial [69] involving cirrhotic patients with sepsis-induced hypotension, 5% human albumin led to better hypotension reversal, improved hemodynamics, and higher short-term survival compared to saline.

The “randomized-controlled trial comparing 20% ALbumin to Plasmalyte in patients with cirrhosis and Sepsis-induced hypotension” (ALPS) [70] compared 20% albumin to balanced crystalloid in cirrhotic patients with sepsis-induced hypotension. Albumin was more effective in achieving the target MAP at 3 h, with a faster decline in arterial lactate levels and a reduced need for dialysis. However, albumin was associated with pulmonary complications, leading to discontinuation in 22% of the albumin-treated patients. The 28-day mortality rate was similar between the groups.

#### 4.1.2. Meta-Analyses of Albumin in Sepsis

A letter to the editor of the *New England Journal of Medicine* highlighted the collective findings of three large-scale studies [71]. These trials, which included SAFE [63], ALBIOS [66], and EARSS [65], collectively indicated a trend towards reduced mortality in sepsis patients receiving albumin with relative risks closely aligning from 0.87–0.94. Although no individual trial reached statistical significance, a pooled analysis suggested a significant reduction in mortality with a relative risk of 0.92, supporting earlier meta-analyses that incorporated smaller trials with varied control fluids and patient demographics [71].

Several meta-analyses have explored the use of albumin-containing solutions in the resuscitation of critically ill patients, yielding varying findings over time. Recent studies, however, suggest that albumin administration is safe, although a definitive survival advantage over crystalloids remains inconclusive in these analyses [13,72,73,74,75,76,77]. There has been a significant reduction in 90-day mortality associated with the use of albumin compared to crystalloids in the context of shock resuscitation [12]. 

## 5. Comparative Analysis

Drawing on a parallel relationship between albumin use in cirrhosis and sepsis, one finds both similarities and differences in their clinical contexts and outcomes. Albumin is well-established in cirrhosis management, effectively addressing complications, such as fluid imbalance, circulatory dysfunction, and infection. It helps in plasma volume expansion, restoring oncotic pressure, and provides anti-inflammatory and antioxidant benefits. The advantages of albumin are less definitive in sepsis. Despite similar inflammatory responses and endothelial dysfunction in sepsis and cirrhosis, their pathophysiology and clinical presentations differ. Albumin is thought to maintain vascular integrity and modulate inflammation in infections. Some studies have pointed to hemodynamic stabilization effects [6], but these have not consistently shown clear clinical benefits. Concerns regarding pulmonary edema and other adverse effects also exist.

### 5.1. Inconsistent Effectiveness

In cirrhosis, the effectiveness of albumin arises from its ability to address specific complications related to hepatic decompensation, particularly post-paracentesis circulatory dysfunction after LVP, HRS-AKI, and SBP. The picture is less clear in patients with more heterogeneity in their cirrhotic presentation, including HE, non-SBP infection, and acute-on-chronic liver failure [5]. In cirrhotic patients with sepsis, trials have shown promising results in terms of hypotension reversal and short-term survival benefits [69,70].

The heterogeneous nature of sepsis, involving various pathogens and clinical presentations, complicates the assessment of the therapeutic efficacy of albumin. Patients vary widely in underlying infections, comorbidities, and clinical presentation., the application of albumin in this context is less straightforward. However, prophylactic or indiscriminate use in sepsis has not consistently demonstrated clear advantages. This variance in the etiology and progression of sepsis adds complexity to the role of albumin.

The selection of treatment strategies that focus on varying immunological responses during sepsis and septic shock should shift towards emphasizing the host’s reaction rather than the pathogen. This change in perspective is particularly relevant given the diverse nature of sepsis, suggesting that a more promising approach may be to address the varied responses of the host’s immune system [78]. 

### 5.2. Organ Dysfunction and Volume Expansion

Cirrhosis leads to portal hypertension, which results in bacterial translocation and exacerbates systemic inflammation, oxidative stress, and immune dysfunction. Albumin administration helps to alleviate these complications by restoring oncotic pressure, reducing inflammation, and stabilizing the circulatory system [32]. In contrast, sepsis involves a dysregulated immune response to infection and widespread endothelial dysfunction [9]. Although albumin may contribute to maintaining vascular integrity, modulating inflammation, and mitigating endothelial dysfunction in sepsis, the clinical manifestations and underlying pathophysiology of sepsis differ significantly from those of cirrhosis.

Given the effectiveness of albumin in HRS-AKI, septic AKI may be the preferential focus of its use in sepsis. Sepsis-associated acute kidney injury (SA-AKI) is notably influenced by hypoalbuminemia as a risk factor and can be mitigated through small-volume resuscitation strategies [79,80]. A recent consensus by Zarbock et al. [81] highlighted the role of albumin in fluid resuscitation in SA-AKI. High-molecular-weight colloids, such as albumin, theoretically cause selective expansion of the intravascular space. However, this effect may be compromised when vascular permeability is altered and the endothelial glycocalyx is damaged during inflammation. The administration of supplemental albumin, as a preferred colloid over synthetic colloids, is recommended if substantial fluid replacement is required. Nonetheless, there are currently no data supporting the routine use of albumin for volume resuscitation in sepsis patients. Furthermore, there are limited data to suggest a cut-off value for crystalloid infusion above which albumin should be considered as part of the resuscitation fluid. 

In managing SA-AKI, particularly with underlying hypoalbuminemia [80,82,83], small-volume resuscitation using hyper-oncotic albumin solutions could offer a balanced approach to maintaining effective circulatory volume with an anti-inflammatory agent without precipitating fluid overload [59], thus potentially reducing the risk and severity of AKI in sepsis.

### 5.3. Disease Severity and Hypoalbuminemia

Hypoalbuminemia, is associated with disease severity [84], and a low serum albumin concentration at the time of discharge is a prognostic factor for 1-year mortality in survivors of sepsis [85]. In decompensated cirrhosis, serum albumin undergoes both structural and functional changes, impacting its non-oncotic properties such as antioxidant scavenging, immune-modulating, and endothelial-protective functions. Consequently, hypoalbuminemia (a quantitative change) and qualitative alterations can substantially reduce the amount of ‘effective’ circulating albumin [32]. Systemic inflammation intensifies with the progression of cirrhosis and becomes more pronounced in decompensated states and acute-on-chronic liver failure. Albumin administration attenuates systemic inflammation in patients with decompensated cirrhosis, as indicated by improved markers of systemic inflammation [47]. Studies have attributed the anti-inflammatory properties of albumin to its uptake by endothelial and immune cells, mediated by its antioxidant properties and ability to block endosomal TLR signaling [86]. This inflammation, characterized by oxidative stress and pro-inflammatory and anti-inflammatory biomarkers, parallels the systemic changes observed in sepsis [78]. However, the complications and outcomes in cirrhotic patients differ from those in patients with sepsis owing to additional factors such as portal hypertension [47].

In the context of sepsis, these findings suggest that albumin may play a crucial role beyond volume expansion. Its immunomodulatory and endothelium-protective effects, along with its ability to attenuate systemic inflammation, highlight the potential of albumin not just as a resuscitative fluid but also as a therapeutic agent targeting the host response to sepsis. The lessons learned from cirrhosis, particularly regarding qualitative changes in albumin and its systemic inflammatory impact, could inform strategies for managing sepsis, especially in cases where hypoalbuminemia and systemic inflammation are prominent.

### 5.4. Dose and Duration of Administration

In the context of decompensated cirrhosis, recent studies have highlighted the significance of albumin dose and duration of administration in modifying disease progression, preventing complications, and improving survival and health care costs. The ANSWER study demonstrated that long-term albumin administration (40 g twice a week for 2 weeks, followed by 40 g/week for 18 months) significantly improved 18-month overall survival, reduced the need for paracentesis, and lowered the incidence of complications such as SBP, AKI, and hepatic encephalopathy [34]. This study contrasts with the MACHT trial, in which a lower dose of albumin (40 g/15 days) was used, and no significant differences in complications or mortality were observed [35]. The key difference between these studies lies in the amount of albumin administered and inclusion of a loading dose in the ANSWER study. This approach led to a significant and sustained increase in serum albumin levels, which was not observed in the MACHT trial. The dose of albumin and its effect on serum albumin levels are crucial for the full development of the biological properties of albumin. A high dose of albumin (1.5 g/kg body weight/week) improved effective volemia and systemic inflammation in patients with stable decompensated cirrhosis, as opposed to lower doses [48].

Extrapolating these findings to sepsis treatment, it is suggested that the dose and duration of albumin administration could be critical factors for its effectiveness. The RASP study on the efficacy of 4% albumin for sepsis in cancer patients found no significant improvement in 7-day survival rates compared to lactated Ringer’s alone [68]. Conducted on 360 cancer patients with severe sepsis or septic shock, the study’s results raise questions about the adequacy of an albumin dosage of less than 100 g. 

In cirrhosis, clinically beneficial disease-modifying effects were observed when the serum albumin level reached 4 g/dL [48]. In contrast, while the ALBIOS trial targeted a serum albumin level of 3 g/dL [66], a similar goal in the SAFE study was rarely met, with only a few patients reaching 3.5 g/dL, primarily those who started with less reduced initial albumin levels [87]. None of these studies showed positive outcomes in terms of mortality endpoints. Therefore, higher doses of albumin or doses adjusted for therapeutic response [88] could potentially improve the effectiveness of albumin in patients with sepsis. 

### 5.5. Safety

The ALPS trial [75] raised safety concerns regarding 20% albumin in patients with cirrhosis and sepsis-induced hypotension. Albumin use has been linked to a decline in the PaO_2_/FiO_2_ ratio, indicating potential volume overload [89]. Adverse effects, including pulmonary edema, allergic reactions, bronchospasm, and rebound hypertension, prompted discontinuation in 22% of patients [70]. The ATTIRE trial [20] focused on 777 patients with chronic liver failure, reporting 169 serious adverse events (SAEs) in 148 patients, with concerns regarding albumin infusions causing pulmonary edema, portal hypertensive complications, or gastrointestinal bleeding.

Pneumonia was the most frequent infection in the trial participants [20,70], but its role as a risk factor for albumin-related pulmonary edema is unclear. Potential risk factors may include cirrhosis-related systemic inflammation that affects the heart or lungs. Both trials highlighted the need for cautious albumin use while monitoring for volume overload and pulmonary edema.

Bleeding is another concern, especially in patients with sepsis who are prone to coagulation abnormalities [90]. Albumin administration might increase bleeding risk due to hemodilution, particularly in patients with pre-existing coagulopathies or active bleeding [91]. Increased bleeding complications were observed in the albumin group in the ATTIRE trial [20]. The “ALBumin In Cardiac Surgery” (ALBICS) trial [92] linked 4% albumin use in cardiac surgery to increased perioperative bleeding, necessitating more blood transfusions and re-sternotomies [19]. These findings indicate that while albumin aids in volume expansion, it may increase bleeding risk, which is a factor to consider in sepsis treatment and other clinical scenarios.

## 6. Clinical Implications and Future Directions

Albumin’s established role in cirrhosis, effectively managing complications like LVP, AKI, HRS-AKI, and SBP, contrasts with its emerging but complex role in sepsis, where treatment is complicated by the disease’s heterogeneity [48,62]. In cirrhosis-related sepsis, albumin demonstrates potential for managing hypotension and offering short-term survival benefits [69,70], particularly in SA-AKI where hypoalbuminemia is prevalent [80,81].

Figure 2 underscores the importance of albumin dosage and duration in therapy, a topic that has gained prominence in recent studies. The differential impact of albumin dosage in cirrhosis and sepsis treatment indicates the need for continued research to determine the optimal treatment protocols in sepsis.

Safety concerns, particularly risks of pulmonary edema and bleeding complications, highlighted in trials such as ALPS and ATTIRE, remain a major consideration [20,70]. These risks emphasize the need for cautious albumin use and monitoring, especially in vulnerable patient populations. The relationship between albumin administration, pneumonia, and pulmonary complications in the context of cirrhosis and sepsis is complex and requires further investigation.

Future research should aim to explore these complex relationships further, focusing on the optimal use of albumin in sepsis treatment. This includes understanding patient-specific responses, dosage optimization, and assessing long-term outcomes. The figure sets the stage for future studies to explore the mechanistic pathways of albumin’s impact on sepsis-related complications, paving the way for more tailored and effective treatment strategies.

## 7. Conclusions

This narrative review clarifies the role of albumin in sepsis, based on its proven efficacy in cirrhosis-related complications. Albumin in cirrhosis effectively addresses fluid imbalance, circulatory dysfunction, and inflammation, particularly in the management of post-paracentesis circulatory dysfunction, HRS-AKI, and SBP. In sepsis, the role of albumin is less clear and complex owing to the heterogeneity of sepsis. Although it may maintain vascular integrity and modulate inflammation, its consistent benefits in sepsis outcomes have not been confirmed. Safety concerns such as pulmonary edema and bleeding necessitate caution. Albumin may be more beneficial for specific sepsis subgroups, such as those with hypoalbuminemia, hypervolemia, advanced liver disease, or kidney failure. Its use in restrictive resuscitation protocols with balanced crystalloids, aiding diuretic resistance in de-resuscitation, and stabilizing hemodynamics in kidney replacement therapy is notable.

Practical recommendations for clinicians include personalized albumin therapy for sepsis tailored to individual patient needs based on clinical presentation and conditions. Addressing hypoalbuminemia in sepsis is crucial, given the promising results in cirrhosis with higher on-treatment serum albumin levels. Clinical trials are necessary to clarify the role of albumin in sepsis, evaluate its effectiveness in correcting hypoalbuminemia, assess patient outcomes, and determine the optimal dosages and protocols.

## Figures and Tables

**Figure 1 ijms-24-17606-f001:**
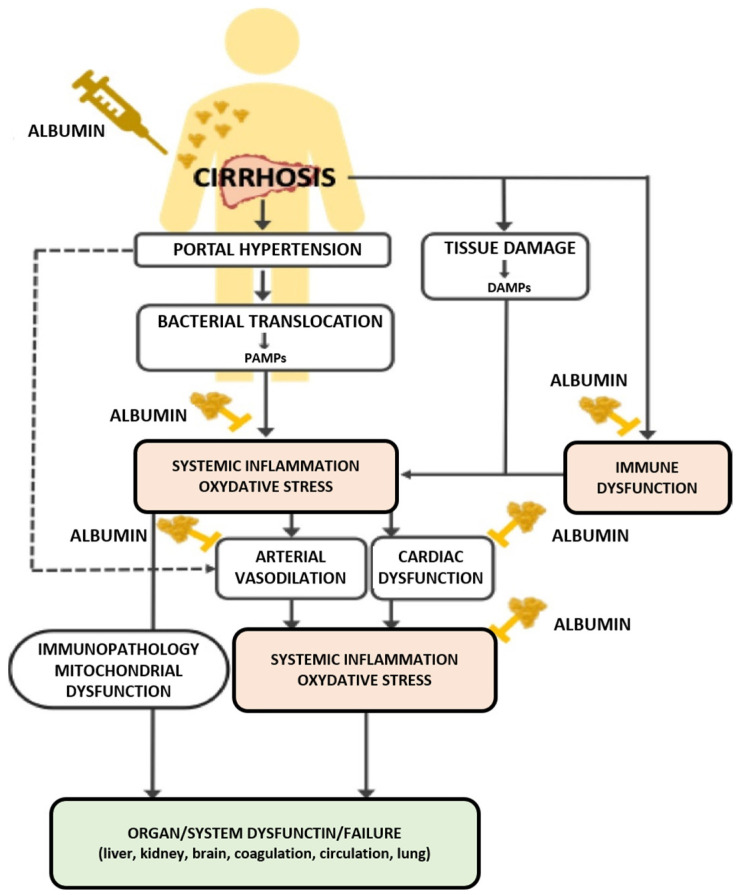
Pathophysiology of decompensated cirrhosis and potential positive effects of albumin administration. PAMPs: pathogen-associated molecular patterns, DAMPs: damage-associated molecular patterns. Reproduced without modification from Pompili et al. 2023 [5] (accessed on 8 November 2023), under a Creative Commons Attribution-NonCommercial-NoDerivatives 4.0 International (CC BY-NC-ND 4.0) License (https://creativecommons.org/licenses/by-nc-nd/4.0/ (accessed on 8 November 2023). Copyright © 2023, The Authors. This reuse has not been endorsed by the licensor. The source reference is “[5]” and is available at https://www.ejinme.com/article/S0953-6205(23)00232-7/fulltext (accessed on 8 November 2023).

**Figure 2 ijms-24-17606-f002:**
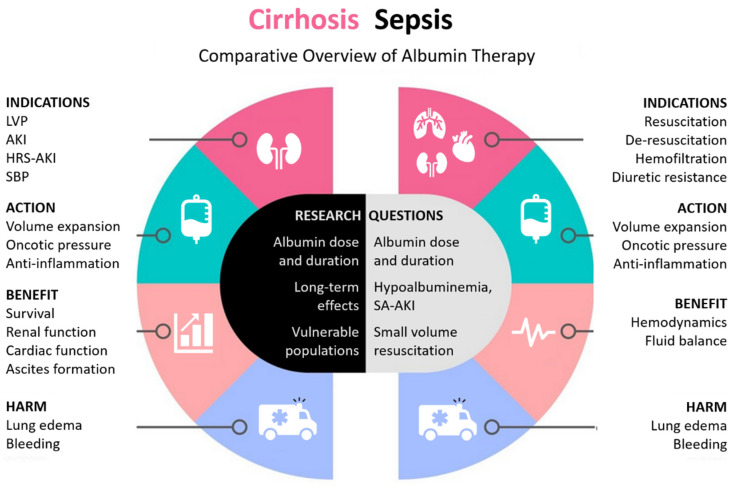
Comparative Analysis of Albumin Therapy in Cirrhosis vs. Sepsis. Specific indications and actions of albumin in cirrhosis and sepsis are outlined, along with their clinical benefits and potential harms. Central to the graphic are shared research questions that address dosage and impact, signaling directions for future study. LVP, large-volume paracentesis; AKI, acute kidney injury; HRS, hepatorenal syndrome; SBP, spontaneous bacterial peritonitis; SA, sepsis-associated. Designed using assets from Freepik.com.

**Table 1 ijms-24-17606-t001:** Key Insights from Cirrhosis on Albumin Infusion Effectiveness and Safety.

Lessons	Key Insights
Inconsistent Effectiveness	Varies by cirrhosis scenario
	Less pronounced in LVP, non-SBP infections, and overt HE
	Short-term use aligns with colloid properties of albumin
Organ Dysfunction and Volume Expansion	Highly effective in decompensated cirrhosis requiring volume expansion
	Beneficial for SBP and HRS-AKI, reducing renal failure and mortality
Disease Severity and Hypoalbuminemia	Serum albumin levels inform prognostic scoring
	Disease severity influences albumin dose needs
	Short-term use mainly effective in SBP or HRS-AKI
Dose and Duration of Administration	Higher doses/longer durations improve outcomes
	Reduced mortality and inflammation with long-term use
Safety	Requires careful dosing to prevent complications
	Risks include pulmonary edema and bleeding in vulnerable patients

Abbreviations: HRS-AKI, hepatorenal syndrome-acute kidney injury; LVP, large-volume paracentesis; SBP, spontaneous bacterial peritonitis.

**Table 2 ijms-24-17606-t002:** Summary of studies on albumin vs. crystalloids in sepsis resuscitation.

Study	Year	Patient Population	Intervention	Control	Dose and Duration of Albumin	Primary Endpoint	Outcomes
SAFE [63,64]	2004	Critically ill	4% Albumin	0.9% Saline	All fluid resuscitation in ICU until death, discharge, or for up to 28 days; volumes reducing over time (2247 mL on days 1 to 4)	Mortality, Organ Failure, ICU Stay	Similar outcomes at 28 days; post hoc reduced 28-day mortality and reduced vasopressor use in sepsis subgroup
EARSS [65]	2011	Septic shock	20% Albumin	0.9% Saline	100 mL every 8 h for 3 days	28-Day Survival	No significant mortality difference
ALBIOS [66,67]	2014	Severe sepsis or septic shock	20% Albumin	Crystalloids	Administered 300 mL after randomization, from day 1 until day 28 or ICU discharge titration of albumin serum levels to 3 g/dL	90-Day Survival	Improved MAP and reduced heart rate; shorter time to cessation of vasopressor or inotropic agents; no difference in new organ failures; no difference in mortality; post hoc reduced 90-day mortality in septic shock subgroup
RASP [68]	2019	Septic cancer patients in ICU	4% Albumin	Lactated Ringer’s	Mean volume of 2402 mL during initial 6 h	Mortality at day 7 and day 28	No significant improvement in mortality rates
FRISC [69]	2021	Cirrhotic patients with sepsis-induced hypotension	5% Albumin	0.9% Saline	Administered 250 mL over 15–30 min followed by 50 mL/h for 3 h	Hypotension reversal at 1 and 3 h, survival at one week	Higher rate of hypotension reversal, sustained reductions in heart rate and hyperlactatemia, higher proportion of surviving patients at one week
ALPS [70]	2022	Cirrhotic patients with sepsis-induced hypotension	20% Albumin	Plasmalyte	Administered 0.5–1.0 g/kg over a period of 3 h	MAP > 65 mmHg at 3 h	Albumin more effective, faster decline in lactate, reduced need for dialysis, longer time to dialysis initiation, similar 28-day mortality, associated with pulmonary complications

Abbreviations: ICU, intensive care unit; MAP, mean arterial pressure.

## Data Availability

No new data were created.

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
