# Peer review of "Controversies Surrounding Albumin Use in Sepsis: Lessons from Cirrhosis"

_ijms, 2023, doi:10.3390/ijms242417606_

Round 1

Reviewer 1 Report

Comments and Suggestions for Authors

I read with great interest the manuscript by Wiedermann on the Albumin use in sepsis. The review is sound and novel. However, there are some issues that need to be addressed:

- Line 60-62. Please provide additional background on the use of albumin in patients with cirrhosis in the introduction section to justify the need for this review.

- Table 1. Please summarize the key insight into bullet points in order to increase readability of the table.

- Line 179-182. Please provide a reference for this sentence.

- All the subparagraph should be in bold type to help the reader to see the separation among paragraphs.

- Line 262-264. It should also be underlined that SSC 2021 guidelines recommend the use of continuous assessment to guide fluid therapy and avoid fluid overload. This can be done by inferior vena cava echocardiographic assessment (doi: 10.1186/s40635-023-00529-z - doi: 10.1186/s40635-023-00505-7) or hemodynamic monitoring (doi: 10.1007/s00134-014-3525-z). Please discuss and add these 3 references.

Reviewer 2 Report

Comments and Suggestions for Authors

This narrative review provides an overview of albumin use in sepsis with a special reference to its applications in patients affected by cirrhosis. The aim of the author is to make a sort of parallelism between the use of albumin in cirrhosis and its applications in septic patients, highlighting the main differences, the complications, the pathophysiological roots and possible future implications.

Major comments:

I am a bit concerned on the length of the whole work, and I believe that a summary (especially regarding the paragraph of albumin in cirrhosis) would be beneficial.

Another big concern is related to citations: it is quite understandable that the author as a particular expertise on albumin applications in critically ill patients and AKI, as shown by self-citation. However, I believe that there are a bit too much self-citations. Moreovere, ref.n. 78 and ref.n.80 are the same.

It is difficult to extrapolate information on future directions, even if the author tried to summarize on a table (table 3). I believe that rewriting the last paragraph, with a special emphasis on future perspectives on sepsis studies would be beneficial. Moreover, the table remains difficult to consultation, and a graphical representation of its contents would benefit the whole work.

Paragraph 2 should be summarized and annexed to the introduction, as at the end the author states the rational for the parallelism that is the basis of the review itself.

Even if a narrative review, i would suggest the author to introduce a paragraph explaining the search conducted in order to write the work, and criteria underlying selection of the studies described.

Minor comments:

To keep reading easiness, I suggest the author to avoid the use of acronyms when they are used only once or few times in the text. 

Round 2

Reviewer 2 Report

Comments and Suggestions for Authors

The revised version of the manuscript has improved. I believe that the work should be published in its present version. I applaude the author for the work and the interesting topic analyzed. All the concerns arisen in the previous revision have been fully addressed.